# Constrained GPI for Zero-Shot Transfer in Reinforcement Learning

**Jaekyeom Kim**
Seoul National University
jaekyeom@snu.ac.kr

**Seohong Park**
University of California, Berkeley
seohong@berkeley.edu

**Gunhee Kim**
Seoul National University
gunhee@snu.ac.kr

## Abstract

For zero-shot transfer in reinforcement learning where the reward function varies between different tasks, the successor features framework has been one of the popular approaches. However, in this framework, the transfer to new target tasks with generalized policy improvement (GPI) relies on only the source successor features [6] or additional successor features obtained from the function approximators' generalization to novel inputs [12]. The goal of this work is to improve the transfer by more tightly bounding the value approximation errors of successor features on the new target tasks. Given a set of source tasks with their successor features, we present lower and upper bounds on the optimal values for novel task vectors that are expressible as linear combinations of source task vectors. Based on the bounds, we propose *constrained GPI* as a simple test-time approach that can improve transfer by constraining action-value approximation errors on new target tasks. Through experiments in the Scavenger and Reacher environment with state observations as well as the DeepMind Lab environment with visual observations, we show that the proposed constrained GPI significantly outperforms the prior GPI's transfer performance. Our code and additional information are available at https://jaekyeom.github.io/projects/cgpi/.

## 1   Introduction

For sequential decision making, deep reinforcement learning (RL) has been shown to be effective for various types of problems including games [31] and robotics [20, 22]. With such great successes, interest in multi-task RL has also surged, where its goal is to train a single agent that can efficiently solve multiple varying tasks. In multi-task RL, we focus on the *transfer learning* setting, where the agent learns shared structural knowledge from a set of source tasks during training, and exploits and generalizes them in new, unseen target tasks at test time.

One popular approach to transfer in RL is to leverage the successor features (SFs) framework [1, 6, 7, 12, 25], which transfers policies learned on source tasks to target tasks, where the tasks share the same environment dynamics but differ in their reward functions. Successor features build a representation of value functions decoupled from reward functions, and transfer to the tasks with arbitrary reward functions by taking an inner product with corresponding task vectors. They utilize generalized policy improvement (GPI) [6], which generalizes policy improvement with multiple policies and provides the performance lower bounds for GPI policies.

However, GPI does not take into account any information from the *smoothness* of optimal action-value functions with respect to task vectors. Tackling this issue, Borsa et al. [12] propose universal successor features approximators (USFAs), which can estimate the optimal successor features for novel task vectors. Nevertheless, the function approximator can make high approximation errors on the task vectors, especially when the new task vectors are distant from the source task vectors. For instance, when USFAs are trained with source tasks to get close to given goals, they may not

36th Conference on Neural Information Processing Systems (NeurIPS 2022).

generalize well to the target tasks where the agent should get away from the given goals. That is, if the elements of target task vectors have the opposite signs from the source task vectors, USFAs could output successor features with high approximation errors.

To improve the successor features approximation of USFAs for the new tasks, we aim at bounding value approximation errors on the new target tasks. We first introduce a new theorem on bounding the optimal values for the tasks that are expressible as linear combinations of source tasks. Our theorem generalizes the conical combination condition used by the prior theorem by Nemecek and Parr [25]. Using our new bounds as constraints, we can train the successor features approximators whose action-value approximation errors on novel tasks are bounded. We extend this idea so that we accomplish a similar effect with no additional training; as a result, we propose *constrained GPI* as a test-time approach to bounding the approximation errors. Despite its simplicity and no need for modification to the training procedure, we empirically show that constrained GPI attains large performance improvements compared to the original GPI in multiple environments, including the Scavenger [8, 9] and Reacher [13] environments with state observations and the DeepMind Lab [7, 10, 12] environment with first-person visual observations.

Our main contributions can be summarized as follows:

- We present a novel theorem on lower- and upper-bounding optimal values for novel tasks that can be expressed as *linear* combinations of source tasks. It extends and generalizes the previous theorem for *conical* combinations by Nemecek and Parr [25], to enable a broader application of the bounds.

- Based on our new theorem, we propose *constrained GPI* as a simple test-time approach that can improve transfer to novel tasks by constraining action-value approximation errors on new target tasks, with no modification to the training procedure.

- We empirically show that our approach can improve the performance over the GPI baselines by large margins in the Scavenger, Reacher and DeepMind Lab environments. We also provide analyses for a better understanding of our results.

## 2 Preliminaries

We describe the problem setting and background on successor features and universal successor features approximators. We refer the reader to Appendix for an in-depth discussion of related work.

### 2.1 The Zero-Shot Transfer Problem in RL

We define a Markov Decision Process (MDP) as $M \equiv (\mathcal{S}, \mathcal{A}, P, R, \gamma)$. $\mathcal{S}$ and $\mathcal{A}$ are the state and action spaces, respectively. $P(\cdot|s, a)$ defines the transition probability distribution of the next states given $s \in \mathcal{S}$ and $a \in \mathcal{A}$. $R(s, a, s')$ is the reward for taking action $a$ at state $s$ resulting in $s'$, and $\gamma \in (0, 1]$ is the discount factor. We assume that rewards are bounded.

We consider the zero-shot transfer problem; as in [6], each task is defined by its task vector $\boldsymbol{w} \in \mathbb{R}^d$, and only the reward functions differ across tasks, being decomposed as

$$R_{\boldsymbol{w}}(s, a, s') = \phi(s, a, s')^\top \boldsymbol{w}, \tag{1}$$

where $\phi(s, a, s') \in \mathbb{R}^d$ is the features of $(s, a, s')$. We denote the set of source task vectors as $\mathcal{T}$, which is used for training. At test time, we evaluate the transferred policy on each target task $\boldsymbol{w}' \notin \mathcal{T}$ with no additional update of pre-trained components. We examine both the possible scenarios: (i) the features $\phi(s, a, s')$ are available to the agent [6, 12] and (ii) no pre-defined features are available and the agent needs to construct its own features and task vectors. We first introduce the formulation for (i) in Section 2.2 and then its variant for (ii) in Section 2.3.

## 2.2 Successor Features and Universal Successor Features Approximators

We now review successor features (SFs) [6] and how they are transferred to different tasks. Equation (1) allows expressing the action-value function for policy $\pi$ on task $\boldsymbol{w}$ as

$$Q_{\boldsymbol{w}}^{\pi}(s,a) = \mathbb{E}^{\pi}\left[\sum_{i=0}^{\infty}\gamma^i r_{t+i}\Big| S_t = s, A_t = a\right] = \mathbb{E}^{\pi}\left[\sum_{i=0}^{\infty}\gamma^i \phi_{t+i}\Big| S_t = s, A_t = a\right]^{\top}\boldsymbol{w} = \psi^{\pi}(s,a)^{\top}\boldsymbol{w},$$

(2)

where $\phi_t = \phi(s_t, a_t, s_{t+1}) \in \mathbb{R}^d$. Here, $\psi^{\pi}(s,a) \in \mathbb{R}^d$ is called the SFs for policy $\pi$ at $(s,a)$, and taking its inner product with an arbitrary task $\boldsymbol{w}$ results in the action-value for $\pi$ on $\boldsymbol{w}$; *i.e.*, $Q_{\boldsymbol{w}}^{\pi}(s, a)$. Thanks to the analogy between (rewards $r$, action-value functions $Q$) and (features $\phi$, successor features $\psi$), the Bellman equation applies to SFs and thus they can be trained similarly to the way action-value functions are learned; *e.g.*, Q-learning.

The policy improvement theorem [11] states that a new policy that takes a greedy action according to a given policy's value function at each state performs at least as well as the original policy. Generalized policy improvement (GPI) [6] extends policy improvement to the case where the value functions of multiple policies are available. Given a task $\boldsymbol{w}'$, a set of policies $\pi_1, \ldots, \pi_n$, their action-value functions $Q_{\boldsymbol{w}'}^{\pi_1}, \ldots, Q_{\boldsymbol{w}'}^{\pi_n}$ and their approximations $\tilde{Q}_{\boldsymbol{w}'}^{\pi_1}, \ldots, \tilde{Q}_{\boldsymbol{w}'}^{\pi_n}$, the GPI policy is defined as

$$\pi_{\text{GPI}}(s) \in \operatorname*{argmax}_{a} \max_{i} \tilde{Q}_{\boldsymbol{w}'}^{\pi_i}(s,a).$$

(3)

Barreto et al. [6] suggest that $Q_{\boldsymbol{w}'}^{\pi_{\text{GPI}}}(s,a) \geq \max_i Q_{\boldsymbol{w}'}^{\pi_i}(s,a) - \frac{2}{1-\gamma}\max_i \left\|Q_{\boldsymbol{w}'}^{\pi_i} - \tilde{Q}_{\boldsymbol{w}'}^{\pi_i}\right\|_{\infty}$. They also provide the upper bound on the suboptimality of the GPI policy as

$$\|Q_{\boldsymbol{w}'}^* - Q_{\boldsymbol{w}'}^{\pi_{\text{GPI}}}\|_{\infty} \leq \frac{2}{1-\gamma}\left\{\min_i \|\phi\|_{\infty}\|\boldsymbol{w}' - \boldsymbol{w}_i\| + \max_i \left\|Q_{\boldsymbol{w}'}^{\pi_i} - \tilde{Q}_{\boldsymbol{w}'}^{\pi_i}\right\|_{\infty}\right\},$$

(4)

where each $\pi_i$ is an optimal policy for $\boldsymbol{w}_i$.

While the GPI theorem allows the transfer of learned successor features to arbitrary tasks that share the same environment dynamics, it is limited in the following aspect. GPI uses the action-values for source tasks on target tasks based on the reward decomposition assumption (Equation (1)) *i.e.*, $\tilde{Q}_{\boldsymbol{w}'}^{\pi_i}(s,a) = \tilde{\psi}^{\pi_i}(s,a)^{\top}\boldsymbol{w}'$ for each $i$. However, it does not take any advantage of the *smoothness* of the optimal action-value functions with respect to different task vectors [12].

To overcome this limitation, Borsa et al. [12] introduce universal successor features approximators (USFAs). Inspired by universal value functions (UVFs) [30], they extend the original successor features with policy vectors $\boldsymbol{z} \in \mathbb{R}^l$ as input to their approximators. More specifically, universal successor features (USFs) are defined to satisfy

$$\psi^{\pi_{\boldsymbol{z}}}(s,a) \equiv \psi(s,a,\boldsymbol{z}) \approx \tilde{\psi}(s,a,\boldsymbol{z}),$$

(5)

where $\boldsymbol{z}$ is a policy vector for the policy $\pi_{\boldsymbol{z}}$, and USFAs $\tilde{\psi}$ are the learned approximators of USFs $\psi$. Naturally, the value functions are expressed as

$$\tilde{Q}_{\boldsymbol{w}}^{\pi_{\boldsymbol{z}}}(s,a) = \tilde{\psi}(s,a,\boldsymbol{z})^{\top}\boldsymbol{w} \approx \psi(s,a,\boldsymbol{z})^{\top}\boldsymbol{w} = Q_{\boldsymbol{w}}^{\pi_{\boldsymbol{z}}}(s,a).$$

(6)

Each reward function induces optimal policies, which can be encoded using the corresponding task vectors. That is, one can simply choose to define the policy vector space to be the same as the task vector space ($l = d$) and let $\boldsymbol{z} = \boldsymbol{w}$ be a policy vector of an optimal policy for task $\boldsymbol{w}$. Then, $\pi_{\boldsymbol{w}}$ and $Q_{\boldsymbol{w}}^{\pi_{\boldsymbol{w}}}$ denote an optimal policy for $\boldsymbol{w}$ and its action-value function, respectively.

The training of USFAs is similar to that of SFs, except for that it additionally involves sampling of policy vectors given task vectors. The update of USFAs at the $k$-th iteration is

$$\tilde{\psi}^{(k+1)} \leftarrow \operatorname*{argmin}_{\psi} \mathbb{E}_{\boldsymbol{w}\sim\mathcal{T}, \boldsymbol{z}\sim\mathcal{D}_{\boldsymbol{z}}(\cdot|\boldsymbol{w}), (s,a,s')\sim\mu}\left[\left\|\phi(s,a,s') + \gamma\tilde{\psi}^{(k)}(s',a',\boldsymbol{z}) - \psi(s,a,\boldsymbol{z})\right\|^2\right]$$

(7)

for $a' = \operatorname{argmax}_b \tilde{\psi}^{(k)}(s',b,\boldsymbol{z})^{\top}\boldsymbol{z}$. $\mathcal{D}_{\boldsymbol{z}}(\cdot|\boldsymbol{w})$ is the policy vector distribution; for instance, $\mathcal{N}(\boldsymbol{w}, \sigma I)$ can be used for better training with diversified inputs. $\mu$ is the transition sampling distribution, which

involves the GPI policy of the samples from $\mathcal{D}_{\boldsymbol{z}}(\cdot|\boldsymbol{w})$ or a replay buffer. We use gradient descent to update the parameters.

USFAs provide a benefit that they allow a GPI policy to use an arbitrary set of policies $\{\pi_{\boldsymbol{z}}\}_{\boldsymbol{z}\in\mathcal{C}}$ as $\pi_{\text{GPI}}(s) \in \operatorname{argmax}_a \max_{\boldsymbol{z}\in\mathcal{C}} \tilde{Q}^{\pi_{\boldsymbol{z}}}_{\boldsymbol{w}'}(s,a)$. However, the generalization of USFAs to new policy vectors depends on a function approximator $\psi$, and thus if $\mathcal{C}$ contains policy vector(s) distant from source vectors, a GPI policy with $\mathcal{C}$ may have high approximation errors and perform poorly or even worse than a GPI policy with only source vectors [12], as will be demonstrated later in our experiments.

### 2.3 Universal Successor Features Approximators with Learned $\phi$

For the scenario where features $\phi$'s are not provided to the agent[1], we adopt the problem formulation from Ma et al. [23] where for each task the task information $\boldsymbol{g} \in \mathcal{G}$ is given to the agent. Although the task information $\boldsymbol{g}$, unlike a task vector, cannot be directly combined with successor features for transfer to a novel task, zero-shot inference could still be possible by leveraging the information about the task.

Specifically, we not only perform the original learning of $\tilde{\psi}$ letting the task information induce policy vectors instead, but also train $\tilde{\phi}$ and $\tilde{\boldsymbol{w}}$ to approximate the reward decomposition with transition samples. As done in [23], we update $\tilde{\psi}$, $\tilde{\phi}$ and $\tilde{\boldsymbol{w}}$ using gradient descent to minimize $\mathbb{E}_{\boldsymbol{g}\sim\mathcal{T}^{\boldsymbol{g}},\boldsymbol{z}\sim\mathcal{D}^{\boldsymbol{g}}_{\boldsymbol{z}}(\cdot|\boldsymbol{g}),(s,a,r,s')\sim\mu}\left[\mathcal{L}^{\psi}+\mathcal{L}^{Q}\right]$ for

$$\mathcal{L}^{\psi} := \frac{1}{d}\left\|\tilde{\phi}(s,a,s') + \gamma\tilde{\psi}^{(k)}(s',a',\boldsymbol{z}) - \tilde{\psi}(s,a,\boldsymbol{z})\right\|^2 \tag{8}$$

$$\mathcal{L}^{Q} := \left\{r + \gamma\tilde{\psi}^{(k)}(s',a',\boldsymbol{z})^{\top}\tilde{\boldsymbol{w}}^{(k)}(\boldsymbol{z}) - \tilde{\psi}(s,a,\boldsymbol{z})^{\top}\tilde{\boldsymbol{w}}(\boldsymbol{z})\right\}^2 \tag{9}$$

and $a' = \operatorname{argmax}_b \tilde{\psi}^{(k)}(s',b,\boldsymbol{z})^{\top}\tilde{\boldsymbol{w}}^{(k)}(\boldsymbol{z})$ at the $k$-th iteration. The superscript $(k)$ denotes the target, $\mathcal{T}^{\boldsymbol{g}}$ is the source task information set, $\mathcal{D}^{\boldsymbol{g}}_{\boldsymbol{z}}(\cdot|\boldsymbol{g})$ is the policy vector distribution conditioned on the task information and $\mu$ is the sampling distribution.

## 3 Constrained GPI for Improved Zero-Shot Transfer of Successor Features

To mitigate the aforementioned issue of the possibly unlimited approximation errors of USFAs, we propose a simple yet effective method that improves the transfer of successor features by further leveraging the reward decomposition structure in Equation (1). We first present, under a more relaxed condition, lower and upper bounds on the optimal values for novel task vectors that are expressed as linear combinations of source task vectors (Section 3.1). Then, we propose a novel approach called *constrained GPI*, which effectively confines the approximated action-values inside the computed lower and upper bounds (Section 3.2).

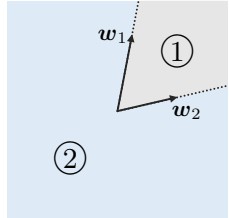

Figure 1: An example comparison of task space coverage. While the conical combinations of $\boldsymbol{w}_1$ and $\boldsymbol{w}_2$ covers only area ①, the linear combinations covers both area ①–②.

### 3.1 Bounding Optimal Values for New Tasks

Theorem 1 of [25] provides the lower and upper bounds on the value of an optimal policy for a new task, whose vector $\boldsymbol{w}'$ is a *positive conical combination* of source task vectors *i.e.*, $\boldsymbol{w}' = \sum_{\boldsymbol{w}\in\mathcal{T}}\alpha_{\boldsymbol{w}}\boldsymbol{w}$ such that $\alpha_{\boldsymbol{w}} \geq 0, \forall \boldsymbol{w}\in\mathcal{T}$ and $\sum_{\boldsymbol{w}\in\mathcal{T}}\alpha_{\boldsymbol{w}} > 0$[2]. However, for a broad application of such bounds, the positive conical combination condition can be too restrictive, since the resulting bounds only apply to the task vectors that appear inside the conical hull of source task vectors.

Therefore, we suggest a more relaxed theorem, which holds for an arbitrary task vector $\boldsymbol{w}'$ that is expressed as a linear combination of the source task vectors *i.e.*, $\boldsymbol{w}' = \sum_{\boldsymbol{w}\in\mathcal{T}}\alpha_{\boldsymbol{w}}\boldsymbol{w}$ for $\alpha_{\boldsymbol{w}} \in \mathbb{R}, \forall \boldsymbol{w}\in\mathcal{T}$. Figure 1 shows an example that compares the task space coverage of conical [25] and

---

[1]One typical example presented later in our experiments is the case where the agent observes visual inputs. Then, it is not trivial to derive features that linearly decompose reward functions.

[2]We slightly abuse the notation and let $\alpha_{\boldsymbol{w}}$ denote the coefficient for vector $\boldsymbol{w}$.

our linear combinations. With our extended task space coverage, we can apply the bounds to more general target tasks outside of the conical hull, which will be further discussed in the next section.

We define $\epsilon_{\boldsymbol{w}_2}^{\pi_{\boldsymbol{w}_1}}$ to be an upper bound on the approximation error of $\tilde{Q}_{\boldsymbol{w}_2}^{\pi_{\boldsymbol{w}_1}}$ for arbitrary tasks $\boldsymbol{w}_1, \boldsymbol{w}_2$ such that

$$|Q_{\boldsymbol{w}_2}^{\pi_{\boldsymbol{w}_1}}(s,a) - \tilde{Q}_{\boldsymbol{w}_2}^{\pi_{\boldsymbol{w}_1}}(s,a)| \leq \epsilon_{\boldsymbol{w}_2}^{\pi_{\boldsymbol{w}_1}}(s,a), \ \ \forall (s,a) \in \mathcal{S} \times \mathcal{A}, \tag{10}$$

and we present our theorem as follows.

**Theorem 1.** *Given a task vector $\boldsymbol{w}' = \sum_{\boldsymbol{w} \in \mathcal{T}} \alpha_{\boldsymbol{w}} \boldsymbol{w}$ for $\alpha_{\boldsymbol{w}} \in \mathbb{R}, \forall \boldsymbol{w} \in \mathcal{T}$, for all state-action pairs $(s,a) \in \mathcal{S} \times \mathcal{A}$, the action-value of $\pi_{\boldsymbol{w}'}$, which is an optimal policy for task $\boldsymbol{w}'$, on task $\boldsymbol{w}'$ is lower- and upper-bounded as $L_{\boldsymbol{w}',\mathcal{T}}(s,a) \leq Q_{\boldsymbol{w}'}^{\pi_{\boldsymbol{w}'}}(s,a) \leq U_{\boldsymbol{w}',\mathcal{T},\boldsymbol{\alpha}}(s,a)$ for*

$$L_{\boldsymbol{w}',\mathcal{T}}(s,a) := \max_{\boldsymbol{w} \in \mathcal{T}} \left[ \tilde{Q}_{\boldsymbol{w}'}^{\pi_{\boldsymbol{w}}}(s,a) - \epsilon_{\boldsymbol{w}'}^{\pi_{\boldsymbol{w}}}(s,a) \right], \tag{11}$$

$$U_{\boldsymbol{w}',\mathcal{T},\boldsymbol{\alpha}}(s,a) := \sum_{\boldsymbol{w} \in \mathcal{T}} \max \left\{ \alpha_{\boldsymbol{w}} \left( \tilde{Q}_{\boldsymbol{w}}^{\pi_{\boldsymbol{w}}}(s,a) + \epsilon_{\boldsymbol{w}}^{\pi_{\boldsymbol{w}}}(s,a) \right), \alpha_{\boldsymbol{w}} C_{\boldsymbol{w}}(s,a) \right\}, \tag{12}$$

*for some $C_{\boldsymbol{w}}(s,a) \leq \min_{\pi} Q_{\boldsymbol{w}}^{\pi}(s,a)$ such as $C_{\boldsymbol{w}}(s,a) = \frac{1}{1-\gamma} r_{\boldsymbol{w}}^{min}$ where $r_{\boldsymbol{w}}^{min}$ is the minimum reward on $\boldsymbol{w}$ i.e., $r_{\boldsymbol{w}}^{min} = \min_{(s,a) \in \mathcal{S} \times \mathcal{A}} R_{\boldsymbol{w}}(s,a)$ and $\boldsymbol{\alpha} = \{\alpha_{\boldsymbol{w}}\}_{\boldsymbol{w} \in \mathcal{T}}$.*

*Proof.* For the derivation of the lower bound $L_{\boldsymbol{w}',\mathcal{T}}(s,a)$, since $Q_{\boldsymbol{w}'}^{\pi_{\boldsymbol{w}'}}$ is the optimal action-value function for task $\boldsymbol{w}'$ and $Q_{\boldsymbol{w}'}^{\pi_{\boldsymbol{w}'}}(s,a) \geq Q_{\boldsymbol{w}'}^{\pi_{\boldsymbol{w}}}(s,a)$ for arbitrary task $\boldsymbol{w}$ and state-action pair $(s,a)$,

$$Q_{\boldsymbol{w}'}^{\pi_{\boldsymbol{w}'}}(s,a) \geq \max_{\boldsymbol{w} \in \mathcal{T}} Q_{\boldsymbol{w}'}^{\pi_{\boldsymbol{w}}}(s,a) \geq \max_{\boldsymbol{w} \in \mathcal{T}} \left[ \tilde{Q}_{\boldsymbol{w}'}^{\pi_{\boldsymbol{w}}}(s,a) - \epsilon_{\boldsymbol{w}'}^{\pi_{\boldsymbol{w}}}(s,a) \right]. \tag{13}$$

For the upper bound $U_{\boldsymbol{w}',\mathcal{T},\boldsymbol{\alpha}}(s,a)$, we use that $Q_{\boldsymbol{w}}^{\pi_{\boldsymbol{w}'}}(s,a) \leq Q_{\boldsymbol{w}}^{\pi_{\boldsymbol{w}}}(s,a)$ and $Q_{\boldsymbol{w}}^{\pi_{\boldsymbol{w}'}}(s,a) \geq \min_{\pi} Q_{\boldsymbol{w}}^{\pi}(s,a) \geq C_{\boldsymbol{w}}(s,a)$ for arbitrary task $\boldsymbol{w}$ and state-action pair $(s,a)$, which leads to

$$Q_{\boldsymbol{w}'}^{\pi_{\boldsymbol{w}'}}(s,a) = \sum_{\boldsymbol{w} \in \mathcal{T}} \alpha_{\boldsymbol{w}} \left( Q_{\boldsymbol{w}}^{\pi_{\boldsymbol{w}'}}(s,a) - C_{\boldsymbol{w}}(s,a) \right) + \sum_{\boldsymbol{w} \in \mathcal{T}} \alpha_{\boldsymbol{w}} C_{\boldsymbol{w}}(s,a) \tag{14}$$

$$\leq \sum_{\boldsymbol{w} \in \mathcal{T}} \max \left\{ \alpha_{\boldsymbol{w}} \left( Q_{\boldsymbol{w}}^{\pi_{\boldsymbol{w}'}}(s,a) - C_{\boldsymbol{w}}(s,a) \right), 0 \right\} + \sum_{\boldsymbol{w} \in \mathcal{T}} \alpha_{\boldsymbol{w}} C_{\boldsymbol{w}}(s,a) \tag{15}$$

$$\leq \sum_{\boldsymbol{w} \in \mathcal{T}} \max \left\{ \alpha_{\boldsymbol{w}} \left( Q_{\boldsymbol{w}}^{\pi_{\boldsymbol{w}}}(s,a) - C_{\boldsymbol{w}}(s,a) \right), 0 \right\} + \sum_{\boldsymbol{w} \in \mathcal{T}} \alpha_{\boldsymbol{w}} C_{\boldsymbol{w}}(s,a) \tag{16}$$

$$= \sum_{\boldsymbol{w} \in \mathcal{T}} \left\{ \max \left\{ \alpha_{\boldsymbol{w}} \left( Q_{\boldsymbol{w}}^{\pi_{\boldsymbol{w}}}(s,a) - C_{\boldsymbol{w}}(s,a) \right), 0 \right\} + \alpha_{\boldsymbol{w}} C_{\boldsymbol{w}}(s,a) \right\} \tag{17}$$

$$= \sum_{\boldsymbol{w} \in \mathcal{T}} \max \left\{ \alpha_{\boldsymbol{w}} Q_{\boldsymbol{w}}^{\pi_{\boldsymbol{w}}}(s,a), \alpha_{\boldsymbol{w}} C_{\boldsymbol{w}}(s,a) \right\} \tag{18}$$

$$\leq \sum_{\boldsymbol{w} \in \mathcal{T}} \max \left\{ \alpha_{\boldsymbol{w}} \left( \tilde{Q}_{\boldsymbol{w}}^{\pi_{\boldsymbol{w}}}(s,a) + \epsilon_{\boldsymbol{w}}^{\pi_{\boldsymbol{w}}}(s,a) \right), \alpha_{\boldsymbol{w}} C_{\boldsymbol{w}}(s,a) \right\}. \tag{19}$$

$$\square$$

In Equation (19), for each $\boldsymbol{w} \in \mathcal{T}$, the sign of $\alpha_{\boldsymbol{w}}$ determines which of the two terms in the $\max$ operator is used. If $\alpha_{\boldsymbol{w}} \geq 0$, the $\max$ operator selects the first term, whereas a negative $\alpha_{\boldsymbol{w}}$ lets the second term be used. Note that our Theorem 1 recovers Theorem 1 of [25] when $\boldsymbol{w}'$ is a conical combination of $\boldsymbol{w}$'s from $\mathcal{T}$ i.e., $\alpha_{\boldsymbol{w}} \geq 0, \forall \boldsymbol{w} \in \mathcal{T}$.

Intuitively, this theorem states the condition that the optimal action-value for an arbitrary target task must satisfy, given the optimal successor features for the source tasks. The theorem is applicable to different problems wherever bounding of optimal values is useful. One example is policy cache construction, where the agent should decide whether to reuse existing policies in the cache set or learn a new one given each new task [25]. As will be shown in the next section, we employ the bounding as a constraint on the action-values for novel target tasks, for the guidance of transfer. In Sections 4.1 and 4.2 and appendix B, we empirically show that the application of our Theorem 1 can significantly improve the performance in the cases where target tasks are outside the conical hull of source tasks.

## 3.2 Constrained Training and Constrained GPI

As described in Section 2.2, the universal successor features approximators (USFAs) [12] improve the original successor features so that arbitrary policy vectors, including the ones for target tasks, can be used for GPI. However, the use of arbitrary policy vectors with USFAs solely relies on the generalization power of the approximators (*e.g.*, neural networks). Thus, the obtained successor features on novel tasks might contain high approximation errors, which could make the GPI policy perform poorly.

Our high-level idea to tackle the issue is to exploit the reward decomposition structure in Equation (1) even for obtaining SFs for new tasks, instead of solely relying on the approximators. We employ the lower and upper bounds on optimal values from Theorem 1 to enforce the bounds on the approximate successor features. As a result, the approximation errors can be reduced by restricting the estimated optimal values to be inside those bounds around the optimal values, which can prevent the use of erroneous values during the transfer to unseen tasks.

For now, we will first introduce how to train the successor features approximators to output the successor features that satisfy the bounds on novel tasks. Then, we will point out that an analogous effect can be accomplished by modifying only the inference algorithm, and propose *constrained GPI* as a simple yet effective *test-time* approach to improving zero-shot transfer to novel tasks.

**Constrained training of SF approximators**. In the original training of USFAs, the approximators are learned with a set of source tasks $\mathcal{T}$ in Equation (7). We propose to guide the training by employing Theorem 1; we impose constraints for the approximators using the lower and upper bounds on the optimal values for arbitrary linear combinations of source tasks. Specifically, for the training of USFAs, we use Equation (7) but with the following constraints:

$$L_{\boldsymbol{w}',\mathcal{T}}(s,a) \leq \tilde{\psi}(s,a,\boldsymbol{w}')^{\top}\boldsymbol{w}' \leq U_{\boldsymbol{w}',\mathcal{T},\xi(\boldsymbol{w}',\mathcal{T},s,a)}(s,a) \quad \text{for } \boldsymbol{w}' \in \mathcal{W}, \tag{20}$$

where $(s,a)$ is the same sample as the main objective of Equation (7). $\mathcal{W}$ is a set of task vectors for the constraints, which can be independent of the source task set $\mathcal{T}$, and $\xi(\cdot)$ determines the coefficients $\boldsymbol{\alpha}$ given a target task $\boldsymbol{w}'$ and $\mathcal{T}$. We will explain later how to determine $\xi(\cdot)$. $\mathcal{W}$ can be any subset of the linear span of source task vectors, but practically, we randomly sample a number of vectors from the span at each update. Since the targets of the constraints are not fixed with respect to both $\boldsymbol{w}'$ and $(s,a)$ throughout the training, we use penalty terms (or soft constraints) that linearly penalize the constraint violations as

$$\frac{1}{|\mathcal{W}|}\sum_{\boldsymbol{w}'\in\mathcal{W}}\Big( \big\{ L_{\boldsymbol{w}',\mathcal{T}}(s,a) - \tilde{Q}_{\boldsymbol{w}'}^{\pi_{\boldsymbol{w}'}}(s,a) \big\}_{+} + \big\{ \tilde{Q}_{\boldsymbol{w}'}^{\pi_{\boldsymbol{w}'}}(s,a) - U_{\boldsymbol{w}',\mathcal{T},\xi(\boldsymbol{w}',\mathcal{T},s,a)}(s,a) \big\}_{+} \Big),$$

where $\{x\}_{+}$ denotes $\max\{x,0\}$.

The constrained training suggested above can make the approximators comply with the bounds for any tasks without requiring any additional interactions with the environment. However, it has some downsides. First, since it is a new training procedure, existing pre-trained models cannot be used. It requires some additional computational cost compared to the naive training of successor features approximators. Second, the enforcement of the constraints for training can introduce additional hyperparameters (*e.g.*, the weight coefficient for the penalty terms). Thus, suboptimal hyperparameters may introduce either instability in the training or a decrease in the performance.

**Test-time constrained GPI**. Our idea starts with the observation that in the constrained training, the learned successor features from source tasks are considered the "trustworthy" features for the constraints, because the USFAs are trained on the source tasks. Besides, only the source successor features are used for computing the constraints for all the other tasks. It implies that the learning of the source successor features better not be affected by other criteria, and more accurate source successor features would produce better constraints for other tasks with smaller errors.

Based on the implication, we propose *constrained GPI*, which can not only overcome the limitation of USFAs as done by the aforementioned constrained training but also have two additional practical merits: (i) it is computationally simpler, and (ii) it is a test-time approach with no training. Simply put, we propose replacing the usual GPI policy with the constrained GPI policy as

$$\pi_{\mathrm{CGPI}}(s) \in \underset{a}{\mathrm{argmax}}\,\underset{\boldsymbol{z}\in\mathcal{C}}{\max}\left[ \min\Big\{ \max\big\{ \tilde{Q}_{\boldsymbol{w}'}^{\pi_{\boldsymbol{z}}}(s,a), L_{\boldsymbol{w}',\mathcal{T}}(s,a) \big\}, U_{\boldsymbol{w}',\mathcal{T},\xi(\boldsymbol{w}',\mathcal{T},s,a)}(s,a) \Big\} \right], \tag{21}$$

where the target task $\boldsymbol{w}'$ is expressible as a linear combination of the source tasks and $\xi(\cdot)$ again outputs $\boldsymbol{\alpha}$ given $\boldsymbol{w}'$ and $\mathcal{T}$ as in Equation (20). $\mathcal{C}$ is a set of policies that we can freely choose when applying the constrained GPI.

The constrained GPI policy selects the actions that maximize the maximum action-values as the original GPI policy does but also caps the values with the lower and upper bound constraints derived from the source successor features. The upper bound constraint fixes the overestimation of values computed with approximate successor features for either the target task $\boldsymbol{w}'$ or any other tasks used for constrained GPI. The lower bound constraint ensures that action-values on the target task for the greedy action selection are at least as close to the optimal target action-values as the lower bounds.

The approximation error terms in the lower and upper bounds *i.e.*, $\epsilon_{\boldsymbol{w}'}^{\pi_{\boldsymbol{w}}}(s,a)$ and $\epsilon_{\boldsymbol{w}}^{\pi_{\boldsymbol{w}}}(s,a)$ in Theorem 1 could be ignored in practice, as long as the approximation errors of the source successor features are sufficiently small. Also, we can obtain the tightest upper bound by defining $\xi(\cdot)$ as

$$\xi(\boldsymbol{w}',\mathcal{T},s,a) \coloneqq \underset{\{\alpha_{\boldsymbol{w}}\}_{\boldsymbol{w}\in\mathcal{T}}}{\operatorname{argmin}}\ U_{\boldsymbol{w}',\mathcal{T},\{\alpha_{\boldsymbol{w}}\}_{\boldsymbol{w}\in\mathcal{T}}}(s,a) \quad \text{subject to } \boldsymbol{w}' = \sum_{\boldsymbol{w}\in\mathcal{T}} \alpha_{\boldsymbol{w}}\boldsymbol{w}. \tag{22}$$

The objective $U_{\boldsymbol{w}',\mathcal{T},\{\alpha_{\boldsymbol{w}}\}_{\boldsymbol{w}\in\mathcal{T}}}(s,a)$ is the sum of the piecewise linear functions. Thus, Equation (22) can be solved with linear programming.

We observe that using the lower bound constraint with $L_{\boldsymbol{w}',\mathcal{T}}(s,a)$ is equivalent to including the successor features for source tasks in the input to the constrained GPI; *i.e.*, $\mathcal{T} \subseteq \mathcal{C}$. Also, since $L_{\boldsymbol{w}',\mathcal{T}}(s,a) \leq U_{\boldsymbol{w}',\mathcal{T},\xi(\boldsymbol{w}',\mathcal{T},s,a)}(s,a)$, there would be no difference between GPI and constrained GPI when $\mathcal{C} = \mathcal{T}$. Thus, in our experiments, we mainly use $\mathcal{C} = \{\boldsymbol{w}'\}$, which is equivalent to using $\mathcal{C} = \mathcal{T} \cup \{\boldsymbol{w}'\}$.

## 4  Experiments

### 4.1  Scavenger Experiments

We start our experiments in the Scavenger environment [8, 9], which can assess our approach with minimal influence from external causes. In Scavenger, the agent is positioned at one of the cells in a $G \times G$ grid, and the goal is to maximize the return by collecting objects. Both the agent and objects are spawned at random locations, and there are $d$ classes of objects where the class determines the value of the reward. The state space is $\mathcal{S} = \{0,1\}^{G \times G \times (d+1)}$, where the first $d$ channels describe the current locations of the objects on the map and the last channel specifies the walls where the agent cannot go and objects do not appear. There are four actions available: $\mathcal{A} = \{\texttt{UP}, \texttt{DOWN}, \texttt{LEFT}, \texttt{RIGHT}\}$, and the agent picks up an object by visiting the cell of the object, which spawns a new object of a random class at a random location. The feature $\phi(s,a,s') \in \{0,1\}^d$ is a one-hot vector whose element represents whether the agent picks up an object of that type or not within the transition. The task vector $\boldsymbol{w} \in \mathbb{R}^d$ determines the reward values for the $d$ different classes of objects. Please see Barreto et al. [9] for the full details.

We evaluate the zero-shot transfer performance of different approaches *i.e.*, we first train USFAs as proposed in [12], and measure the performance of GPI and constrained GPI policies that use the same set of USFAs on target tasks with no further policy updates. We set $G = 11$ and use 20 objects in total with the different numbers of classes; $d = 2$ and $d = 4$. With $d = 4$, we also test the USFAs that are learned with the constrained training for a comparison. We use the standard basis vectors of $\mathbb{R}^d$ as the set of source tasks as done in [12], and evaluate agents on the set of target tasks defined by $\{-1,1\}^d$. Therefore, all the target tasks except for the all-ones vector $\mathbf{1}$ are not covered by the conical hull of source tasks, which requires Theorem 1 for bounding of values.

We train eight USFAs agents for 1M steps, and evaluate them on each target vector 10 times with a fixed set of 10 random seeds. To be invariant to the reward scale differences between different tasks, we normalize the scores (or returns) from the environment by the minimum and maximum scores with respect to all the agents' evaluation episodes on each task.

Figures 2 and 3 compare the performance of the USFAs agents with GPI and constrained GPI for exploitation, following the evaluation scheme suggested by [2]. Although they use the same set of trained USFAs, the constrained GPI brings a notable performance improvement in comparison with the original GPI. Also, Figure 3 suggests that the constrained GPI, the test-time method, can match

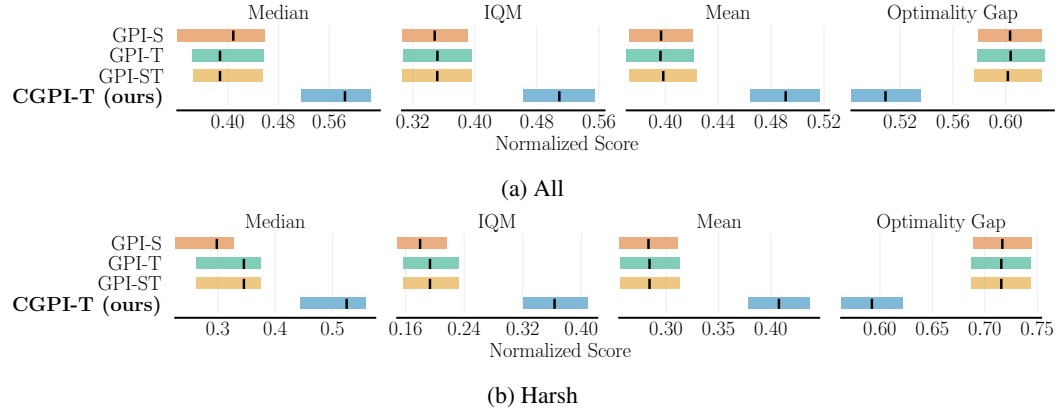

(a) All

(b) Harsh

Figure 2: The aggregated performance metrics with 95% bootstrap confidence intervals [2, 16] of different test-time approaches on Scavenger with $d = 2$. CGPI represents our *constrained GPI*, whereas GPI is the original GPI. The suffixes -S, -T and -ST denote using the set of source task vectors, the target task vector and both as $\mathcal{C}$, respectively. (a) All is the evaluation on the entire set of target task vectors from $\{-1, 1\}^d$, whereas (b) Harsh denotes the evaluation on a subset consisting of 'harsh' tasks, whose number of $-1$'s is no less than that of 1's (*e.g.*, $\boldsymbol{w} = (-1, 1)$).

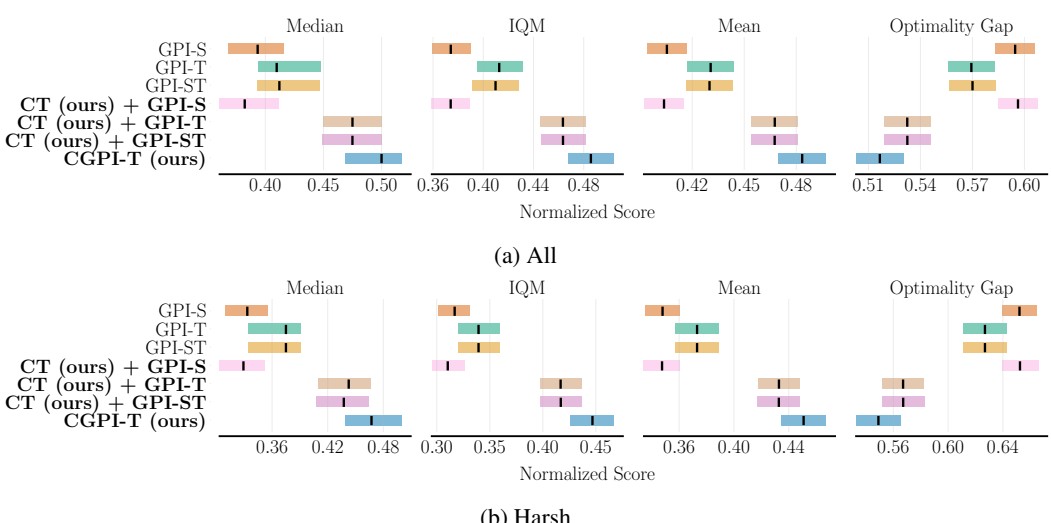

(a) All

(b) Harsh

Figure 3: The aggregated performance metrics with 95% bootstrap confidence intervals [2, 16] of different test-time approaches on Scavenger with $d = 4$. CT denotes the constrained training. We refer the reader to Figure 2 for the full description.

or even outperform the agents learned with the constrained training. One possible explanation is that the constrained training might experience some instability in learning depending on the choice of the hyperparameters, as described in Section 3.2.

In the first and the second columns of Table 1, we present the proportions of the action-values that are changed by the lower and upper bounds of the constrained GPI, measured for the evaluation on Scavenger. The third column shows the proportions of resulting greedy actions changed by them. It implies that USFAs *i.e.*, the function approximators of successor features, may not satisfy the optimal value bounds presented in Theorem 1, and applying the bounds could change a fair proportion of greedy actions to improve the performance.

Table 1: The first and second columns show the proportions of the maximum action-values changed by constrained GPI's lower and upper bounds, during the evaluation on Scavenger. The final column is the proportions of resulting actions changed by them.

| Setting | Lower-bounding | Upper-bounding | Action change |
|---------|----------------|----------------|---------------|
| $d = 2$ | 23.94% | 43.01% | 22.11% |
| $d = 4$ | 5.60% | 46.86% | 20.78% |

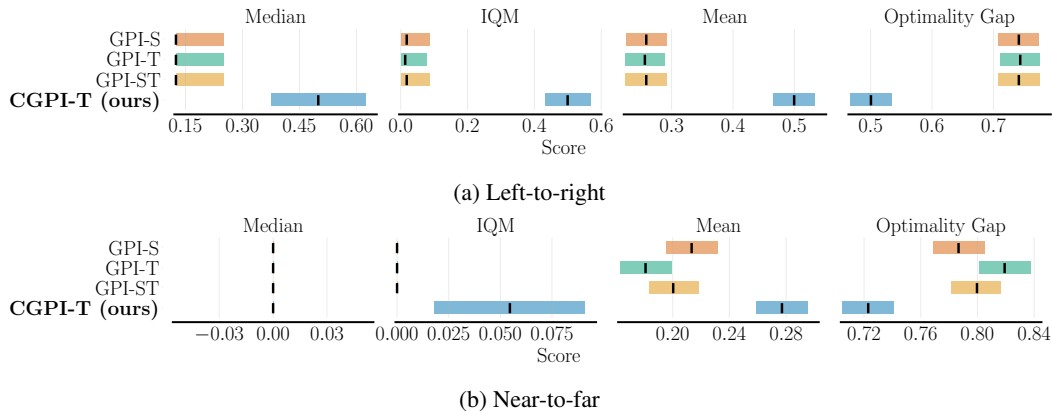

(a) Left-to-right

(b) Near-to-far

Figure 5: The aggregated performance metrics with 95% bootstrap confidence intervals [2, 16] of different test-time approaches with the two scenarios in the DeepMind Lab environment. `CGPI` represents *constrained GPI*. The suffixes `-S`, `-T` and `-ST` denote using the source task information set, the target task information and both for $\mathcal{C}$, respectively. (a) In the left-to-right setting, we train the agent for the goals from the left half and test for the right half. (b) In the near-to-far setting, we train the agent for nearby goals and test for farther goals.

## 4.2 DeepMind Lab Experiments with Learned $\phi$

For evaluation of our approach in a more complex and realistic setting, we employ DeepMind Lab [7, 10, 12] and conduct experiments in a first-person view 3D environment. In a single room, a goal object is placed arbitrarily, and the objective is to reach the goal before the episode ends where its location changes between tasks. Figure 4 shows an example scene that the agent sees with the goal object in red.

At every time step, the agent observes an $84 \times 84 \times 3$ image from the environment and outputs one of $45$ possible actions, which include 5, 3 and 3 choices for `LOOK_LEFT_RIGHT_PIXELS_PER_FRAME`, `STRAFE_LEFT_RIGHT` and `MOVE_BACK_FORWARD` controls, respectively. Since observations are in the first-person view, the goal object may not be seen by the agent, which makes transfer given the task information $\boldsymbol{g}$ critical to the success of the tasks. In each task, we divide the room into an $11 \times 11$ grid and place the goal object in one of the cells. The task information $\boldsymbol{g}$ is a two-dimensional vector that contains the coordinate of the goal in the grid. Starting at the center of the room, the agent receives a reward of one if it reaches the goal within the episode horizon or no rewards otherwise. Therefore, the reward functions are sparse.

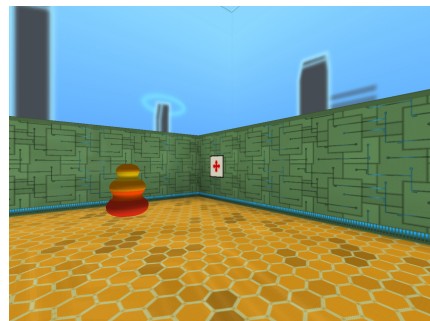

Figure 4: An example scene that the agent sees in the DeepMind Lab tasks. The object is the goal.

For these experiments where the agent observes rendered images rather than the underlying states, it may not be viable to define features $\phi$'s and task vectors $\boldsymbol{w}$'s that linearly decompose reward functions. Therefore, we train agents with the learning of $\tilde{\phi}$ and $\tilde{\boldsymbol{w}}$ from samples from the source tasks with $d = 2$ as described in Section 2.3.

Inspired by Hong et al. [18], we examine zero-shot transfer with the GPI and constrained GPI using two transfer settings: "left-to-right" and "near-to-far". In the "left-to-right" setting, the agent is trained on the source tasks whose goals are sampled from the left half of the room and is tested on the target tasks with goals from the right half. In the "near-to-far" setting, the source tasks have the goals within an $L^\infty = 2$ distance from the center of the room, and target tasks set the goals farther than $L^\infty = 2$.

For each setting, we train eight USFAs agents with different seeds for 3M environment steps on the source tasks and test them on the target tasks. Figure 5 presents the comparison of the GPI with

different $\mathcal{C}$'s and the constrained GPI. Leveraging the same set of trained USFAs with learned $\tilde{\phi}$ and $\tilde{w}$, the constrained GPI outperforms the GPI with the three $\mathcal{C}$'s in both settings by a notable margin. Another observation is that the trained USFAs agents seem to overfit more to the source tasks in the "near-to-far" setting compared to the "left-to-right" setting. It makes the performance on the target tasks much worse. Nonetheless, the constrained GPI is still helpful in such overfitting situations.

## 5    Conclusion and Discussion

We presented constrained GPI, a simple yet effective test-time approach for transfer with approximate successor features. We first focused on the issue that although universal successor features approximators (USFAs) exploit the smoothness of optimal values across different tasks, their approximation errors on novel target tasks could be large especially when those tasks are quite distant from source tasks. Thus, we introduced a theorem about lower and upper bounds on the optimal values for novel task vectors that belong to the task vector space linearly spanned by the set of source task vectors, relaxing the conical combination condition used for the theorem by Nemecek and Parr [25]. We proposed a constrained training scheme making use of those bounds for reducing the action-value errors of the learned approximators on novel tasks. We then suggested *constrained GPI* that uses the bounds at test time to achieve an analogous effect, allowing the use of previously trained models. We empirically showed that this test-time approach can improve the zero-shot transfer performance by a large margin in multiple environments.

**Limitations and future directions**. There may be some cases where the minimum rewards for source tasks *i.e.*, $r_{\boldsymbol{w}}^{\min}$'s are overly small, which could lead to less changes of both action-values and behaviors induced by the upper-bounding in Theorem 1 with $C_{\boldsymbol{w}}(s, a) = \frac{1}{1-\gamma} r_{\boldsymbol{w}}^{\min}$. An interesting direction to tackle the issue is to learn the minimum action-value function during the training and to use the approximate minimum value at each state-action pair as $C_{\boldsymbol{w}}(s, a)$ for deriving the upper bound in Theorem 1. It may allow computing upper bounds more tightly and adaptively for different state-action pairs. Also, if the learned successor features approximators have large errors even on source tasks, not only GPI but also constrained GPI's bounding may not be meaningfully helpful. One idea to mitigate the issue is to take the uncertainty in the approximators and the approximation error term that appears in Theorem 1 into account, *e.g.*, by using ensemble models. As an intriguing direction for future research, we could extend our constrained GPI to other non-linear forms of reward or value decompositions. It may also be interesting to make transfer with successor features and constrained GPI compatible with large-scale approaches for generalization such as [28]. We do not see direct negative societal impacts of this work.

## Acknowledgements

We thank the anonymous reviewers for their valuable comments. This work was supported by Samsung Advanced Institute of Technology, and Institute of Information & communications Technology Planning & Evaluation (IITP) grants funded by the Korea government (MSIT), including (No.2019-0-01082, SW StarLab), (No.2022-0-00156, Fundamental research on continual meta-learning for quality enhancement of casual videos and their 3D metaverse transformation), and (No.2021-0-01343, Artificial Intelligence Graduate School Program (Seoul National University)). Jaekyeom Kim was partly supported by Google PhD Fellowship. Gunhee Kim is the corresponding author.

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
