# A  Related Work

Transfer in reinforcement learning aims at solving a new target task with no additional learning or sample-efficiently by exploiting agents and information obtained from source tasks. We review a line of research with relevant approaches.

**Transfer by reusing policies**. This group of approaches reuses policies learned on source tasks for target tasks. Fernández and Veloso [17] suggest an exploration strategy for the learning of a new policy given a new task and learned source policies, where the gain of using each policy is estimated together on-line and one of the policies in the set is selected probabilistically at each step, based on the gain, but they focus on aiding the training of the target policy with samples from the target task rather than improving the zero-shot transfer performance. On the other hand, Dayan [14] introduce successor representations (SRs), state space occupancy representations disentangled from rewards, which allow linear decomposition of value functions. Barreto et al. [6] propose successor features (SFs) and using SFs as an extension of SRs. Especially, in addition to SFs, which can be combined with arbitrary task vectors for obtaining the corresponding values, Barreto et al. [6] also suggest generalized policy improvement (GPI), allowing composition of multiple source policies on a single task where the resulting GPI policy performs at least as well as any of the source policies. The transfer with SFs and GPI and a number of connected methods [7, 9] combine source policies using the reward decomposition structure but are limited in that they do not further make use of the smoothness of the optimal value functions with respect to different tasks *i.e.*, given two similar tasks, their values are likely to be close to each other. Nemecek and Parr [25] maintain a set of policies and determine whether to learn a new one for a given task or exploit existing ones based on their optimal value bounds, which is a different problem from ours, and they also target tasks expressible as conical combinations of source tasks, whereas we target linear combinations. Alver and Precup [5] suggest sets of assumptions and conditions under which a group of basis policies can induce GPI policies that maximizes undiscounted returns on novel tasks and an iterative algorithm for constructing the basis, but thus they tackle a different problem, many of whose requirements do not apply to more general settings or our problem, due to *e.g.*, (non-binary) continuous reward features $\phi$, no guarantee on possible trajectories, stochastic transition dynamics, etc. Alegre et al. [4] deal with the policy set construction problem as well by interpreting the SFs framework as the learning of multiple policies in the multi-objective RL problem and extending the optimistic linear support algorithm [29] for the SFs framework. Differently from the problem we are tackling, in the maximum entropy setting Hunt et al. [19] aim at compositing source policies for target tasks optimally by estimating and correcting the divergence between source policies, but they consider target tasks whose rewards are convex combinations of source task rewards and the divergence estimation becomes significantly harder when there are more than two source tasks [19].

**Transfer with function approximation**. There is a series of studies that directly exploits the smoothness of optimal values across tasks with function approximators. Schaul et al. [30] propose universal value functions and their approximators, which incorporate goals into their input, and use the approximators for generalization to novel goals. Inspired by [30], Borsa et al. [12] suggest universal successor features approximators (USFAs), which allow the use of GPI with arbitrary approximate policies by extending the original SFs approximators to take policy vectors as their inputs. However, the generalization to novel or distant policy vectors with the function approximators could result in SFs with large approximation errors. In this work, we tackle this problem by leveraging the reward decomposition structure for bounding estimated values around optimal values and thus their errors. On the other hand, Hong et al. [18] propose bilinear value networks (BVN) with a bilinear decomposition of value functions into goal-agnostic and goal-specific components for better sample efficiency in multi-goal reinforcement learning. They provide an alternative formulation of value function approximators differently from the decomposition used in the SFs framework, and thus our approach of bounding approximate values at test time is orthogonal to BVN.

Compared to the existing work mentioned above in general, our constrained GPI is a simple test-time approach for target tasks which are expressible as linear combinations of source tasks.

# B  Robotic Locomotion Experiments

To evaluate our approach in a physical domain, following [6, 25], we employ Reacher, a MuJoCo's robotic locomotion environment [32] from OpenAI Gym [13]. Reacher simulates a robotic arm

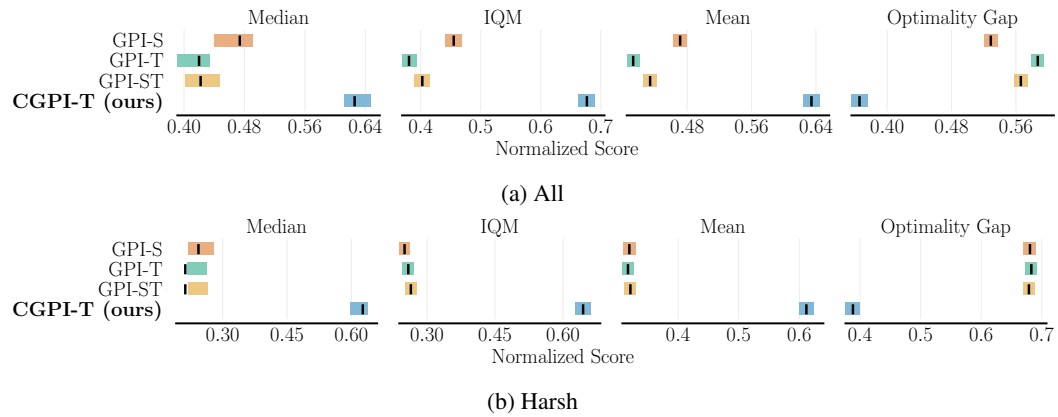

Figure 6: The aggregated performance metrics with 95% bootstrap confidence intervals [2, 16] of different test-time approaches on Reacher with $d = 4$. CGPI represents *constrained GPI*. The suffixes -S, -T and -ST denote using the set of source task vectors, the target task vector and both as $\mathcal{C}$, respectively. (a) All is the evaluation on the entire set of target task vectors from $\{-1, 1\}^d$, whereas (b) Harsh denotes the evaluation on a subset consisting of 'harsh' tasks, whose number of $-1$'s is no less than that of $1$'s.

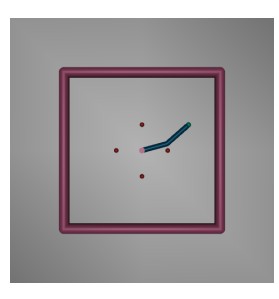

Figure 7: An example scene of Reacher. The four red dots indicate the four goal locations.

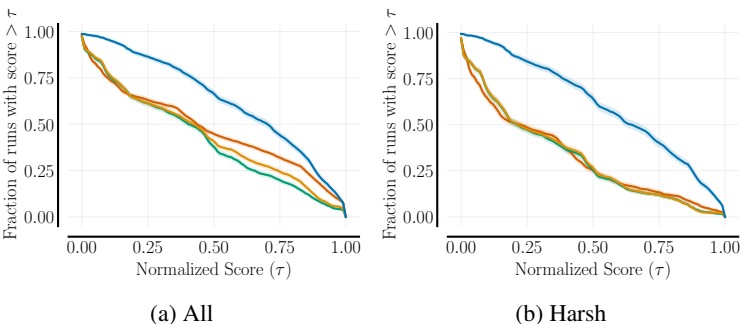

Figure 9: The performance profiles [2, 15] of the inference with GPI and constrained GPI on Reacher. The categories *i.e.*, (a) All and (b) Harsh, and the colors match the ones in Figure 6, and thus the blue lines represent constrained GPI.

with two joints, and its state space $\mathcal{S}$ is 11-dimensional. Its original action space is continuous and two-dimensional, which represents torques at the two hinge joints, and we discretize each action dimension into three values resulting in nine discrete actions in total, as in [6].

For its use in the zero-shot transfer problem, we first set four fixed goal locations at $(0.1, 0.0), (0.0, 0.1), (-0.1, 0.0), (0.0, -0.1)$, and define $\phi(s, a, s') \in \mathbb{R}^4$ to be a vector whose elements are the negative distances between the agent's fingertip and the four goals. USFAs agents are trained with the four standard basis vectors as source tasks, learning to reach or get close to one of the four goals on each of the four source tasks. However, for the evaluation, we define $\{-1, 1\}^d$ to be the set of target tasks. A negative value at each dimension implies not only that the target vector is outside the conical hull of the source tasks but also that the agent would obtain higher rewards with respect to that dimension by getting away from the corresponding goal, instead of reaching the goal. This can make this zero-shot transfer problem challenging, as it requires the agent to do very different behaviors suddenly at test time. For better understanding, Figure 7 shows a rendered scene of Reacher, where the four red dots represent the goal locations.

In our experiments, we train 16 USFAs agents for 1M environment steps to obtain statistically more meaningful results. We evaluate each of the trained agents with 10 episodes for each target vector, again with a fixed set of 10 environment random seeds. Similarly to the Scavenger experiments in the previous section, to take into account the difference in reward scales between different target tasks,

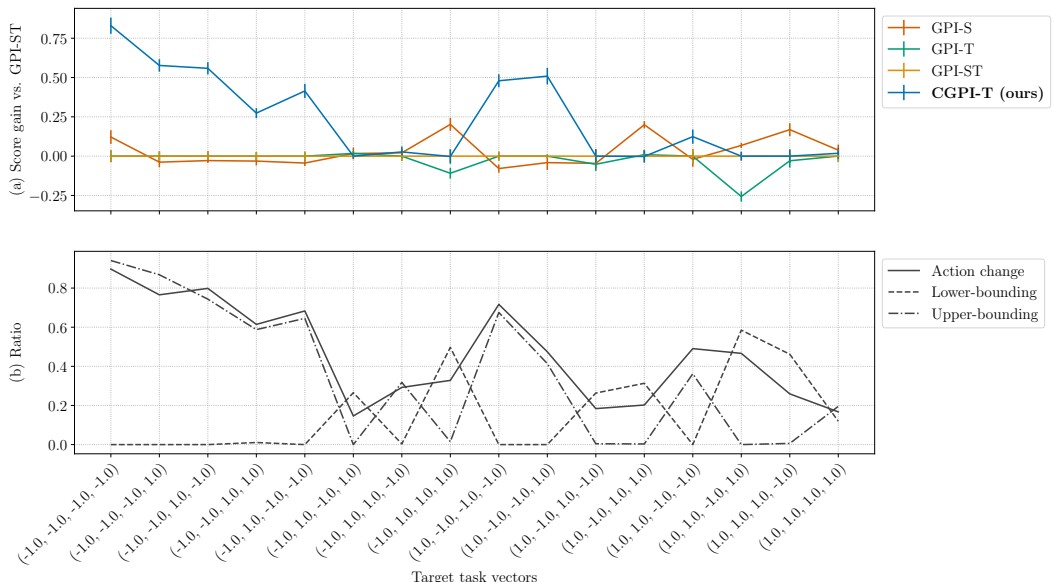

Figure 10: The performance gains of different test-time approaches and the ratios of action and action-value changes due to constrained GPI, with respect to the target task vectors on Reacher. `CGPI` represents *constrained GPI*. The suffixes `-S`, `-T` and `-ST` denote using the set of source task vectors, the target task vector and both as $\mathcal{C}$, respectively. (a) presents the interquartile means (IQMs) of their normalized scores, visualized as performance gains (or differences) compared to `GPI-ST`, where the error bars are the 95% confidence intervals. `Action change` in (b) shows the average portion of the final actions changed by constrained GPI for each target task, during the evaluation. `Lower-bounding` and `Upper-bounding` in (b) denote the average ratios of the action-values after taking the maximums changed by constrained GPI's lower and upper bounds, respectively.

for each of the target tasks, we normalize the returns *i.e.*, scores using the minimum and maximum scores that any of the agents achieve during the evaluation.

Figure 6 provides the comparison of GPI and constrained GPI in terms of the normalized scores. As it shows, the inference with constrained GPI significantly outperforms the ones with GPI on the zero-shot transfer problem. Especially, the gap becomes even larger when they are evaluated on the "Harsh" set of target task vectors, which implies that constrained GPI can be helpful for transferring to different types of target tasks outside the conical hull of source tasks. On the other hand, Figure 9 presents the performance profiles [2, 15] of the approaches with the matching evaluation categories and color mapping. It qualitatively suggests that constrained GPI is more likely to achieve higher scores than GPI.

## C  Further Empirical Analyses

We provide further analyses of the experimental results for a better understanding of our approach. Figure 10 visualizes four quantities with respect to the target task vectors, in the Reacher environment. Figure 10a reports the performance gains of the approaches compared to `GPI-ST` in terms of the interquartile means (IQMs) of the normalized scores on each target task. In Figure 10b, `Action change` shows the average ratio of the action changes due to constrained GPI, for each of the target tasks. `Lower-bounding` and `Upper-bounding` are the average portions of the maximum action-values for the inference that are bounded and changed by constrained GPI's lower and upper bounds, respectively. We restate that the USFAs are trained with the four standard basis task vectors *i.e.*, a (positive) one-hot vector for each of the four dimensions of the task vector space, $\mathbb{R}^4$.

Our first observation is that while the transferred agents perform comparably on some tasks, constrained GPI makes significant differences on the others, especially more on the "Harsh" target tasks with many $-1$'s as elements in their task vectors. It implies that the USFAs, the function approximators, might work reasonably on tasks to which the approximations could be extrapolated

similarly as in the source tasks, but they could perform poorly when there is little or no knowledge easily transferable from the source tasks only with the function approximators.

It also coincides with our second observation: examining the two plots in Figure 10 together, there is a tendency that more action changes by constrained GPI usually result in greater performance gains. This suggests that bounding the action-values at test time with constrained GPI for reducing the value approximation errors often has a meaningful effect on the resulting performance, and the highest increases in both the performance gains and the action change ratios are observed on the "Harsh" target tasks. Thus, we infer that the bounds from Theorem 1 could be effective for constraining the approximation errors of USFAs, especially including target tasks outside the conical combinations of source tasks.

Additionally, `Lower-bounding` and `Upper-bounding` in Figure 10b indicate that a large portion of the action changes and thus the performance gains are related to the upper-bounding in the Reacher environment, and the reduction of the USFAs' overestimation with constrained GPI is an important factor of the performance gains, in this case. Depending on the underlying environment and tasks, there can be target tasks where USFAs underestimate the corresponding action-values and the lower-bounding, which is equivalent to performing GPI with the source task set $\mathcal{T}$ included in the input policy set $\mathcal{C}$, could help improve the resulting performance, as well.

## D    Experimental Details

### D.1    The Scavenger Environment

For the Scavenger experiments, we use the environment from [8, 9]. In addition to the environment settings explained in Section 4.1, we use a maximum episode horizon of 50 and a discount factor of 0.9.

### D.2    The Reacher Environment

Regarding the Reacher experiments, we employ the environment from OpenAI Gym [13]. Besides the environment configurations described in Appendix B, we set the maximum episode horizon as 500 and the discount factor as 0.9. Since original states defined for Reacher contain information about target coordinates, we set those coordinates to zeros for our experiments with different tasks.

### D.3    The DeepMind Lab Environment

For the DeepMind Lab experiments, additionally to the experimental setup introduced in Section 4.2, we render 10 frames per second and use an episode horizon of 50 and a discount factor of 0.99. As the tasks we use are sparse-reward tasks, we sample each episode with one source task vector during the training.

### D.4    Universal Successor Features Approximators and Constrained GPI

We employ the autonomous learning library [26] and PyTorch [27] for the implementation of USFAs. The following selection of hyperparameters was done based on the performance of the USFAs on source tasks, because we have no access to target tasks during the training in the transfer problem and the learning on the source tasks is important for the transfer with both GPI and constrained GPI.

For both of the Scavenger and Reacher environments, we use an MLP with the ReLU nonlinearity and two hidden layers whose sizes are $(128, 256)$, which is chosen over $(64, 128)$ and $(256, 256)$, as USFAs. We set the number of output heads of the networks to the respective number of discrete actions; four for Scavenger and nine for Reacher. For the training of the USFAs, we make use of the Adam optimizer [21] with a learning rate of $1e-4$, which is selected out of $\{3e-4, 1e-4, 3e-5\}$. The target update frequency is set to 100 for Scavenger and 500 for Reacher, where we consider $\{100\}$ for Scavenger and $\{100, 500\}$ for Reacher, and the update frequency is configured as 1 for the two environments. On Scavenger, we use 256-sized replay buffers (chosen out of $\{1, 128, 256\}$) and mini-batches with a size of 1 after testing $\{1, 4, 8, 16\}$. For Reacher, we make replay buffers and mini-batches 2048-sized and 32-sized, which are chosen from $\{2048\}$ and $\{1, 8, 32\}$. For the exploration, we use the $\epsilon$-greedy with its value of 0.1.

For the DeepMind Lab environment, we employ the frame stacking strategy and the network architecture for our shared feature extractor for $\tilde{\psi}$ and $\tilde{\phi}$ from Mnih et al. [24], where the latent feature dimensionality is $64$. $\tilde{w}$ has one hidden layer with $16$ units. A learning rate of $1e-5$ for the Adam optimizer is chosen out of $\{1e-4, 1e-5\}$. The target update frequency and the update frequency are $500$ and $1$, respectively. We use a replay buffer whose size is $65536$ and sample $16$-sized mini-batches, which is chosen from $\{16, 64\}$.

Specifically for USFAs, for the Scavenger and Reacher environments, we sample five policies from $\mathcal{D}_z(\cdot|w)$ at each step, where each dimension of a policy vector is sampled from a Gaussian distribution with a standard deviation of $0.1$ and a mean of the corresponding element of $w$. We use the same number of policy vectors and policy vector distribution but conditioned on the task information instead of task vector, for the DeepMind Lab environment.

For the constrained training of USFAs, we use the same set of hyperparameters, as well as a weight coefficient for the constraints of $0.1$ (chosen from $\{0.1, 1.0, 5.0\}$), where one task is uniformly randomly sampled from the bounded linear span of source tasks at each step. Also, for a more stable learning, we load the model checkpoints of USFAs at the $0.5$M-th step and train for the remaining $0.5$M steps. We universal employ cvxpylayers [3] as a general LP solver for the upper bound with the default solver configurations, for both constrained GPI and the constrained training.

For the Scavenger and Reacher experiments, we conduct the experiments with our CPU machines, where majority of the CPUs are Intel Xeon Gold 6130 or Intel Xeon E5-2695. For each training run, we use two CPU cores without any GPUs for about 6-12 hours. For the DeepMind Lab experiments, we employ our GPU machines and run experiments for about 48-72 hours.