# OpenReview forum: "Constrained GPI for Zero-Shot Transfer in Reinforcement Learning"
_NeurIPS.cc/2022/Conference — NeurIPS 2022 Accept_

### Official Review · Reviewer_Jub8 · 2022-07-10

**Rating:** 6
**Confidence:** 4
**Soundness:** 3 good
**Presentation:** 3 good
**Contribution:** 2 fair

**Summary:**

Successor Features / Generalized Policy improvement provide a method for generalizing to new tasks that are a linear combination of existing tasks. USFA combine this approach to generalization with neural networks with also generalize by training function approximators conditioned on the task.

This work extends USFAs by showing that one can establish lower and upper bounds on the generalization performance. When estimating action-values on a new task, these bounds are used to constrain the function approximator estimate.

They test this approach (comparing to USFAs) on two simple environments and find using these constraints outperforms USFAs.


**Questions:**

Could you comment on the scalability of this approach to larger domains and tasks?


**Limitations:**

Yes although some comments (and ideally testing) on scaling to more complex domains would be useful.

**Strengths And Weaknesses:**


Strengths:

- Paper is well communicated.

- Paper will be of interest to community working on successor features and related approaches.

- Paper introduces both theoretical grounding and empirical results.

Weaknesses:

- In common with a lot of work in SF, the test environments are "toy" and its unclear whether these approaches scale to more interesting tasks. For this reason, it may be of interest only to a smaller subset of the NeurIPS community.

- It would be interesting to compare very different approaches, such as a conditional decision transformer [1] or GATO style policy distillation [2].

- [3] introduces an alternative method for improving the performance of GPI (in the max-ent RL framework) that reduces the generalization error and should be cited, and ideally compared against.

[1] Chen, Lili, et al. "Decision transformer: Reinforcement learning via sequence modeling." Advances in neural information processing systems 34 (2021): 15084-15097.

[2] Reed, Scott, et al. "A generalist agent." arXiv preprint arXiv:2205.06175 (2022).

[3] Hunt, Jonathan, et al. "Composing entropic policies using divergence correction." International Conference on Machine Learning. PMLR, 2019.

---

> ### Author Response · Authors · 2022-08-01
> **Author Response to Reviewer Jub8**
>
> We appreciate the reviewer’s thoughtful and constructive feedback.
>
> **1. Scalability to more interesting tasks**
>
> Although our results include the Scavanger environment with a high-dimensional state space (up to 605 dimensions), we agree with the review that more complex tasks may need to be tested for better evaluation of both successor feature-based methods and our constrained GPI. We are currently preparing additional experimental results on more diverse environments. please see our response to Reviewer Efp4 as well.
>
> **2. Compare with different approaches (Decision Transformer, GATO, Divergence Correction)**
>
> Thank you for the suggestion. Although we fail to obtain empirical results for the suggested methods due to the limited time of rebuttal period, we have revised our related work and conclusion sections (Appendix B and Section 5) with further comparisons and discussions regarding the works.
>
> We appreciate the reviewer’s time and effort put into improving our paper.

---

> > ### Comment · Reviewer_Jub8 · 2022-08-06
> > **Response to author's comments**
> >
> > Thanks for your response. Sounds reasonable and I understand there is limited time during rebuttal for all experiments.
> >
> > I'm probably doing something silly, where do I find the appendix. The pdf seems to just be the main paper, references and checklist (ends at page 12) unless I'm missing something?

---

> > > ### Author Response · Authors · 2022-08-06
> > > **Author Response to Reviewer Jub8**
> > >
> > > Thank you for the additional comment and for understanding the time limitation in the rebuttal period.
> > >
> > > For our appendix, please check out the PDF file attached as the supplementary material, which is placed right below the abstract.

---

### Official Review · Reviewer_drJg · 2022-07-11

**Rating:** 7
**Confidence:** 3
**Soundness:** 3 good
**Presentation:** 4 excellent
**Contribution:** 3 good

**Summary:**

The paper looks at the problem of transfer in reinforcement learning from source tasks to target tasks when the reward signal changes across tasks but the state and action spaces remain the same. Authors address a limitation of a current approach in this setting: GPI with universal successor features approximators (USFAs). The works argues that USFAs can make high approximation errors on test targets if their solutions are distant from the source targets' ones, and proposes a solution to mitigate this limitation.

The solution involves constraining the approximation action-value error on new target task, using a lower and upper bounds introduced in the paper. The experiments show a zero-shot transfer improvement in performance compared to baselines (USFA + GPI) on synthetic scavenger tasks and robotic locomotion tasks.

**Questions:**



Questions
- Is there a way to extend constrained GPI if the reward cannot be linearly decomposed (but an arbitrary non linear function of the features)?
- The theoretical and empirical results presented rely on the prior knowledge of the feature vector $\phi$. How would constrained GPI perform with an unperfected (e.g. learned) feature vector instead?

Minor clarifications and comments:
- It would be helpful to spell out the different methods acronyms of figures 2 and 3 in the main text or in the caption.
- I did not understand caption of figure 2b

**Limitations:**

Very limited discussions on the limitations. In fact, I did not really understand the remark on not getting the "full effectiveness".

No negative societal impacts were reported, and I don't see any myself either.

**Strengths And Weaknesses:**

Strengths:

- The paper addresses relevant and important open problem in RL (transfer learning).
- The writing is clear, rigorous and the paper is easy to follow. A pleasant read!
- I found theoretical contribution to be main significant result, extending Nemecek and Parr result to about the value of an optimal policy for a new task beyond the conical hull is an important result.
- The proposed method, constrained GPI, is an elegant way of improving GPI.

Weaknesses:
The limitations of the work are not really discussed, and no direction for future work is given. This make it difficult to gauge if the method and its guarantees would extend to more complex and large scale environments.

---

> ### Author Response · Authors · 2022-08-01
> **Author Response to Reviewer drJg**
>
> We appreciate the reviewer’s thoughtful and constructive feedback.
>
> **1. Limited discussions on the limitations and future work**
>
> In our original discussion on the limitations, we described that there might be cases where the minimum rewards ($r_w^{\text{min}}$) in source tasks are too small. In such scenarios, actual modification of approximate action-values with our upper bound, which partially contains the minimum reward term, might happen less often, although we empirically find that the upper-bounding does affect the trained agent's values by ~45% and up to ~90% in our Scavenger and Reacher experiments, as shown in Table 1 and Figure 8. As the review points out, another limitation is that the use of successor features needs $\phi$ to be observable or learned with function approximation. Following this suggestion, we have revised Section 5 to elaborate the limitations and future directions of our work.
>
> **2. A way to extend constrained GPI if the reward cannot be linearly decomposed**
>
> Ideally, the successor features (SFs) framework can express arbitrary reward functions by defining some dimensions of $\phi(s, a, s’)$ as the outputs of those reward functions and setting $w$ to a one-hot vector [1, 2]. Nonetheless, as the review suggested, it would be an intriguing future direction to explore more diverse reward decomposition structures (other than the current linear form).
>
> **3. How would constrained GPI perform with an unperfected (e.g. learned) feature vector instead?**
>
> We can train successor features with learned $\phi$ (e.g., [1]). Thus, we expect that CGPI could also be extended to the setting where $\phi$ is not observable, although one caveat is that the increased approximation errors might make the bounds less accurate.
>
> **4. Spell out acronyms in Figures 2 and 3 / Improve the caption of Figure 2b**
>
> We appreciate the suggestion and have updated the captions of Figure 2-4 in the main draft.
>
> Thank you for the time and effort put into improving our paper.
>
> [1] Barreto et al., Successor features for transfer in reinforcement learning, NeurIPS 2017.
> [2] Barreto et al., Transfer in deep reinforcement learning using successor features and generalised policy improvement, ICML 2018.

---

> > ### Comment · Reviewer_drJg · 2022-08-08
> > **Thanks for the response**
> >
> > Thanks for responding and amending the main text.

---

### Official Review · Reviewer_Efp4 · 2022-07-12

**Rating:** 6
**Confidence:** 3
**Soundness:** 4 excellent
**Presentation:** 4 excellent
**Contribution:** 3 good

**Summary:**

The paper notes that Universal successor feature approximators (USFA) $\psi(s,a,z)$ (here $z$ is the policy vector) exploits the smoothness of optimal value functions across different tasks but its approximation error could be large when target task vectors are far away. To improve generalization to a new task $w$ (which can be expressed as a linear combination of source tasks), the paper proposes constrained training of successor features that reduces its approximation error on the new task. It then shows that a similar effect can be achieved by using similar constraints at test time without changing the training of the successor features. The proposed method is able to improve task generalization on scavenger and reacher domains.

**Questions:**

**Recommendation:** I think this is a borderline paper and I am happy to increase my score if the authors can evaluate their method on more tasks (such as the ones in fetch-gym) and compare it to better baselines (such as bilinear value networks).

**Limitations:**

The authors discuss the limitations of their work in Section 5

**Strengths And Weaknesses:**

**Strengths:**
  1. The resulting *constrained GPI* has good theoretical motivations
  2. It exhibits better task generalization than GPI baselines

**Weaknesses:**
  1. The baselines used for comparisons are inadequate. For eg: recent works on bilinear decomposition of Q functions (bilinear value networks (Hong et al., 2022, https://arxiv.org/pdf/2204.13695.pdf)) showed improved generalization on novel tasks. Specifically, for goal condtitioned task with goal $g$, they choose the parameterization $Q(s,a,g) = \phi(s,a)^Tw(s,g)$ (as opposed to $Q(s,a,g) = \phi(s,a,g)^Tw$ or $Q(s,a,g) = \phi(s,a)^Tw(g)$. It would be nice to see comparisons to this bilinear decomposition.
  2. Furthermore, the domains (i.e. scavenger and reacher) on which the method is tested is limited. I would appreciate if authors can provide a more extensive evaluation on various goal conditioned tasks (such as the ones in fetch-gym (https://github.com/jmichaux/gym-fetch)).

---

> ### Author Response · Authors · 2022-08-01
> **Author Response to Reviewer Efp4**
>
> We appreciate the reviewer’s thoughtful and constructive feedback.
>
> **1. Comparison with bilinear value networks (BVN)**
>
> Thank you for suggesting the relevant work. The successor features (SFs) framework has the two main components: (i) the decomposition of reward and value functions ($Q_w(s, a) = \psi(s, a)^\top w$) and (ii) generalized policy improvement (GPI) [1]. We focus on improving GPI with bounding of approximate value functions at test time, while BVN provides a better decomposition of Q-functions. Therefore, we believe our work is orthogonal to BVN (and thus our main baseline is GPI), and applying our constrained GPI to multiple bilinear value functions ($Q(s,a,g)=\psi(s,a)^\top w(s,g)$) is an interesting future research direction. We have included discussions about BVN in the related work section (Appendix B).
>
> **2. The domains on which the method is tested are limited**
>
> We are currently preparing additional experimental results with more diverse environments. It is taking time due to adoption of the environments and evaluation schemes to our codebase and the training from scratch. We will update the draft with extended empirical results by 8th-9th to enable further discussions.
>
> We appreciate the reviewer’s time and effort put into improving our paper.
>
> [1] Barreto et al., Successor features for transfer in reinforcement learning, NeurIPS 2017.

---

> > ### Comment · Reviewer_Efp4 · 2022-08-09
> > **Reply**
> >
> > I thank the author for the response. However, I feel the evaluations for the method is limited to simple tasks/environments and hence I am maintaining my score.

---

> > > ### Author Response · Authors · 2022-08-09
> > > **Author Response to Reviewer Efp4**
> > >
> > > Thank you for the reply.
> > >
> > > Following your suggestion, we conducted additional experiments in the Fetch environment [1] from fetch-gym (<https://github.com/jmichaux/gym-fetch>) with action space discretization for their use within the successor features framework. We used the *VerySparse* version of *Reach*, where the agent receives a reward of 1 when it reaches the target location within a threshold of 0.05 and 0 otherwise. As linear decomposition of this sparse reward function is not trivial, we *learned* $\phi$ and $w$ from the transitions where $w$ determines the task vector given a goal information, similarly to the practice by [2]. We employed the left-to-right setting [3] but with sampling of test-time goals whose coordinate on the left-right axis is 0.05 or more distant from the agent's coordinate, to prevent easy goals from being sampled and make the task more challenging. We trained three agents with different seeds on the training region (*left*) for 1M time steps and tested their zero-shot transfer to a fixed set of 50 goals uniformly randomly sampled from the test region (*right*), using a fixed set of 10 random policy vectors from the training region for both GPI and CGPI. Here are the preliminary results:
> > >
> > > | Zero-shot transfer method | Average return |
> > > | ------------------------- | -------------- |
> > > | GPI w/ source             | 0.080          |
> > > | GPI w/ target             | 0.007          |
> > > | GPI w/ source + target    | 0.080          |
> > > | CGPI (ours)               | 0.213          |
> > >
> > > They suggest that CGPI could be helpful in sparse-reward scenarios such as the Fetch task with no access to the $\phi$  features. We will go beyond these preliminary findings and include the experimental results of CGPI on more tasks with different settings and complexities and their full details in the camera-ready version of our paper.
> > >
> > > We appreciate the reviewer for suggesting the helpful direction to extend our empirical results.
> > >
> > > [1] Plappert et al., Multi-goal reinforcement learning: Challenging robotics environments and request for research, 2018.
> > > [2] Ma et al., Universal successor features for transfer reinforcement learning, 2020.
> > > [3] Hong et al., Bilinear value networks, 2022.

---

> > > > ### Comment · Reviewer_Efp4 · 2022-08-09
> > > > **Re: Author Response to Reviewer Efp4**
> > > >
> > > > I appreciate the author's preliminary results on fetch tasks and I hope that they expand on this initial result in the camera-ready version of the paper. I am increasing my score to 6.

---

### Meta-Review · Area_Chair_PCFb · 2022-08-26

**Recommendation:** Accept
**Confidence:** Certain

**Metareview:**

The reviewers agree that the paper is a valid contribution to the line of research on successor features (SFs) and generalized policy improvement (GPI). They also agree that the paper is well written and easy to follow.

Three points that may be worth taking into account when preparing the final version of the paper, all related to the presentation:

1. The paper has two main contributions: bounds that improve upon those of Nemecek and Parr [1] and a new application of bounds of this type to detect approximation errors in universal successor features approximators (USFAs). Although these two contributions are listed at the end of Section 1, in the rest of the paper the derived bounds are always mentioned in the context of their specific use with USFAs. In particular, I believe the authors never mention that their bounds could also be applied to decide when to add new policies to a set of policies to be used with GPI, as suggested by Nemecek and Parr. In summary, the authors may want to have a presentation that clearly disentangles the two contributions of the paper.

2. Although the writing is mostly clear, I feel like the core idea of the paper is never spelled out in a concise way (this is especially important for the abstract). Given a set of vectors $\mathbf{w}_1$,$\mathbf{w}_2$, $\dots$, $\mathbf{w}_n$, and an associated space of tasks $\mathcal{W}$ composed of all possible linear combinations of the vectors $\mathbf{w}_i$, the paper derives lower and upper bounds for the value functions of all tasks in $\mathcal{W}$ in terms of the value functions of the tasks $\mathbf{w}_i$. These bounds can then be used in several ways; one novel application proposed in the paper is to detect approximation errors in USFAs. Maybe spelling this out in the abstract and introduction would help the reader to quickly understand the message of the paper.

3. It seems like the $\max$ in the upper bound (Eq. 10) will always be resolved based on the sign of $\alpha_{\mathbf{w}}$, since the first term will always be larger when $\alpha_{\mathbf{w}} > 0$ and the second term will always be larger when $\alpha_{\mathbf{w}} < 0$. This seems to be the ``trick'' used to improve upon  Nemecek and Parr's bound. The authors should consider adding this comment to the paper.

I hope the constructive feedback is useful to improve your paper.

[1] Nemecek, M. Parr, R. Policy Caches with Successor Features. ICML, 2021.



**Award:**

No

---

### Decision · Program_Chairs · 2022-09-14

Accept